# Infection and biogeographical characteristics of *Paragonimus westermani* and *P. skrjabini* in humans and animal hosts in China: A systematic review and meta-analysis

Kai Liu[1,☉,‡], Yuan-Chao Sun[1,☉,‡], Rui-Tai Pan[1], Ao-Long Xu[1], Han Xue[1], Na Tian[1], Jin-Xin Zheng[2], Fu-Yan Shi[1]*, Yan Lu[2]*, Lan-Hua Li[1]*

1 School of Public Health, Shandong Second Medical University, Weifang, Shandong, P.R. China, 2 National Institute of Parasitic Diseases, Chinese Center for Disease Control and Prevention; Chinese Center for Tropical Diseases Research; National Key Laboratory of Intelligent Tracking and Forecasting for Infectious Diseases; Key Laboratory on Parasite and Vector Biology, Ministry of Health; WHO Collaborating Centre for Tropical Diseases; National Center for International Research on Tropical Diseases, Ministry of Science and Technology, Shanghai, P.R. China

☉ These authors contributed equally to this work.
‡ These authors share first authorship on this work.
* shifuyan@126.com (FY-S); lyyaner@163.com (YL); orchid8@sina.com (LH-L)

**Data Availability Statement:** All relevant data are within the manuscript and its Supporting Information files.

## Abstract

### Background

Paragonimiasis, primarily caused by *Paragonimus westermani* and *P. skrjabini* in China, is a common food-borne parasitic zoonosis. However, the national distribution of *Paragonimus* spp. infection and its associated environmental determinants remain poorly understood. In this paper, we summarize the infection of *P. westermani* and *P. skrjabini* and describe key biogeographical characteristics of the endemic areas in China.

### Methods

Data on *Paragonimus* infection in humans and animal hosts were extracted from eight electronic databases, including CNKI, CWFD, Chongqing VIP, SinoMed, Medline, Embase, PubMed, and Web of Science. A random-effects meta-analysis model was used to estimate the pooled prevalence. All survey locations were georeferenced and plotted on China map, and scatter plots were used to illustrate the biogeographical characteristics of regions reporting *Paragonimus* infection.

### Results

A total of 28,948 cases of human paragonimiasis have been documented, with 2,401 cases reported after 2010. Among the 11,443 cases with reported ages, 88.05% were children or adolescents. The pooled prevalence of *P. skrjabini* is 0.45% (95% *CI*: 0.27–0.66%) in snails, 31.10% (95% *CI*: 24.77–37.80%) in the second intermediate host, and 20.31% (95% *CI*: 9.69–33.38%) in animal reservoirs. For *P. westermani*, the pooled prevalence is 0.06% (95% *CI*: 0.01–0.13%) in snails, 52.07% (95% *CI*: 43.56–60.52%) in the second

**Funding:** This research was funded by the Shandong Provincial Natural Science Foundation (https://cloud.kjt.shandong.gov.cn/) (ZR2019MH093 to L-H L; ZR2023MH313 to F-Y S), the Shandong Provincial Youth Innovation Team Development Plan of Colleges and Universities (http://edu.shandong.gov.cn/) (2019-6-156 to F-Y S), the National Parasite Resource Bank (https://www.most.gov.cn/index.html) (NPRC-2019-194-30 to Y L), Three-Year Initiative Plan for Strengthening Public Health System Construction in Shanghai (2023-2025) Key Discipline Project (https://wsjkw.sh.gov.cn/) (No. GWVI-11.1-12 to Y L),the Quality Education Teaching Resources Project of Shandong Province and Weifang Medical University (http://edu.shandong.gov.cn/) (SDYAL2022152, 22YZSALK01, and 23YJSALK01 to L-H L), Joint Research Program of China National Center for Food Safety Risk Assessment (https://cfsa.net.cn/) (LH2022GG08 and LH2022GG02 to L-H L), and the National Natural Science Foundation of China (https://www.nsfc.gov.cn/) (81902095 to L-H L). The funders had no role in study design, data collection and analysis, decision to publish, or preparation of the manuscript.

**Competing interests:** The authors have declared that no competing interests exist.

intermediate host, and 21.40% (95% *CI*: 7.82–38.99%) in animal reservoirs. *Paragonimus* are primarily distributed in regions with low altitude, high temperature, and high precipitation. In northeastern China, only *P. westermani* infections have been documented, while in more southern areas, infections of both *P. westermani* and *P. skrjabini* have been reported.

## Conclusions

Paragonimiasis remains prevalent in China, particularly among children and adolescents. Variations exist in the intermediate hosts and geographical distribution of *P. westermani* and *P. skrjabini*. Additionally, altitude, temperature, and precipitation may influence the distribution of *Paragonimus*.

## Author summary

Paragonimiasis, a foodborne zoonotic parasitic disease caused by lung flukes (*Paragonimus* spp.), remains a significant neglected public health threat in many Asian countries, including China. Human infection occurs through the ingestion of raw or undercooked freshwater crab or crayfish containing the metacercariae stage. Given the popularity of consuming raw or undercooked freshwater products in many areas of China, understanding the infection status and spatial distribution of *Paragonimus* spp. in humans and animal hosts is crucial for controlling paragonimiasis. Our study provides a comprehensive summary of the infection levels of the two most important zoonotic *Paragonimus* species, *P. westermani* and *P. skrjabini*, in humans and animal hosts in China, along with a description of the spatial distribution and environmental characteristics of their endemic areas. We observe a wide distribution of *Paragonimus* infection in China, with a significant prevalence found in freshwater crabs and crayfish. Our findings underscore the importance of avoiding the consumption of raw or undercooked freshwater products to prevent foodborne diseases, including paragonimiasis.

## Introduction

Paragonimiasis is a food-borne zoonotic disease caused by several species of lung flukes belonging to genus *Paragonimus* spp. [1]. The infection occurs primarily in the lungs and pleura of humans and animals. When the parasite infects the lungs, it can cause a pulmonary disease resembling tuberculosis and lung cancer [2]. Misdiagnosis of paragonimiasis as pulmonary tuberculosis or lung cancer can lead to significant socioeconomic losses and impose a mental and physical burden on patients due to unnecessary hospitalization, laboratory tests, surgical procedures, and prolonged medication [3]. Human paragonimiasis is widely distributed in Asia, Americas, and Africa, and is still a significant neglected public health threat in China. The global estimate of infected individuals is approximately 20 million, with around 293 million individuals at risk [4]; however, these figures may have been underestimated. New endemic areas are continually being identified, such as in India [5]. It is worth noting that a significant number of paragonimiasis cases have been misdiagnosed as pneumonia, tuberculosis, or lung cancer [6,7]. An estimated 293.8 million individuals are at risk of *Paragonimus* spp. infection, with 195 million of them residing in China [8,9].

More than 30 *Paragonimus* species have been documented in China, among which *P. westermani* and *P. skrjabini* are the most important zoonotic species [2,10]. *P. westermani* (Japanese lung fluke or oriental lung fluke) is most commonly distributed in eastern Asia and in South America, and is the most common cause of human paragonimiasis. *P. skrjabini* is especially prevalent in China, with cases appearing in India and Vietnam as well [1,11]. *P. westermani* followed by *P. skrjabini* are the major pathogens for human paragonimiasis in China [9].

Parasites of *Paragonimus* spp. have a three-host life cycle, with aquatic snails serving as the first intermediate host, freshwater decapod crustaceans as the second intermediate host, while human and other mammals as the definitive host. Human infection is acquired by eating inadequately cooked or pickled freshwater crabs or cray fishes containing the infective forms called metacercariae [12,13]. Drinking untreated stream or river water is also considered to be a possible route of infection [14].

Given the three host nature of the parasite and the fact that consuming raw or undercooked freshwater products is still popular in many areas of China, the infection status of *Paragonimus* spp. in animal hosts is closely related to the epidemic of human paragonimiasis [15]. Therefore, comprehending the level of infection in animals will provide valuable insights for controlling human paragonimiasis. However, prevalence estimates of *Paragonimus* spp. infection in the literature vary greatly across different studies. To date, there has been no comprehensive estimation of *Paragonimus* spp. infection in humans and animal hosts. In addition, very few attempts at the spatial and environmental characteristics of *Paragonimus* spp. infection in China have been made. Consequently, the aims of the current study are to summarize the infection level of two most important zoonotic *Paragonimus* species, *P. westermani and P. skrjabini*, in humans and animal hosts in China, and to describe the spatial distribution and environmental characteristics of their endemic areas.

## Method

### Literature retrieval and selection

This systematic review followed the Preferred Reporting Items for Systematic Reviews and Meta-analyses (PRISMA) reporting guidelines [16], and has been registered with PROSPERO under the identifier CRD42024474528.

A systematic literature search was conducted to identify all studies reporting *Paragonimus* infection in humans and animals from inception to January 1, 2024, using the following electronic databases: China National Knowledge Infrastructure (CNKI), Chinese Wanfang database (CWFD), Chongqing VIP, SinoMed, Medline, Embase, PubMed, and Web of Science. Full-text search was performed using the terms 'paragonimiasis', '*Paragonimus*', 'lung fluke', 'lung trematode', in conjunction with 'China'. The search was limited to English and Chinese languages.

After removing duplicates, two reviewers (KL and YC-S) independently reviewed all the titles and abstracts, with assistance of a third reviewer (RT-P) to reach a consensus in case of disagreement. Subsequently, the full texts were assessed for inclusion by the same reviewers. All studies included in the meta-analysis were published in English or Chinese, and were primary research articles, and epidemiological studies reporting prevalence of *Paragonimus* in humans and animal hosts. Studies were further excluded from meta-analysis if they were letters to the editor, non-epidemiological studies, or had a sample size of fewer than 20 [17]. Additionally, we collected case reports and case series of human infections to summarize the characteristics of cases of human paragonimiasis.

## Data extraction and quality assessment

The following information was extracted from the included articles: title, first author, language, year of publication, year of investigation, study location, *Paragonimus* species, detection method, sample size, number of positive cases, prevalence, taxonomic category of animal hosts (include genus and family), and life style of animal hosts. For human studies, we also collected information on the gender and type of specimens. In population-based surveys, the participants first underwent immunological testing (usually skin testing), and those who tested positive further underwent etiological testing. In this case, the prevalence was calculated using the total number of participants as the denominator, with etiologically confirmed positives as the numerator.

Two reviewers (KL and YC-S) independently evaluated the quality of each included study using a standardized assessment tool developed by Hoy [18]. This tool provides ten items to access the risk of bias, with each item given a score of 0 or 1 for the absence or presence of bias. A summary score of 0–3 indicates a low risk of bias, 4–6 indicates a moderate risk of bias, and 7–10 indicates a high risk of bias.

## Statistical analysis

Freeman-Tukey double arcsine transformation was used to normalize the prevalence and ensure the validity of subsequent analyses [19]. Heterogeneity across studies was assessed using Cochran's Q test and $I^2$ statistics, where $I^2$ statistics quantified the percentage of variation across studies (with $I^2$ values indicating low, moderate, and high heterogeneity at 25%, 50%, and 75%, respectively). If the heterogeneity is statistically significant, a random-effects model was used for meta-analysis; otherwise, a fixed-effects model was used [20,21]. The random-effects model and Peto method were ultimately used to estimate the pooled prevalence as well as their confidence interval (*CI*) in this study [22], following the results of the heterogeneity test. Additionally, subgroup analyses were employed to explore the potential source of heterogeneity across studies, conducted meta-regression analysis with moderators as independent variables and prevalence as the dependent variable to further assess the effects of moderators on the prevalence.

$R^2$, QM, and QE statistics were utilized to interpret the results of subgroup and meta-regression analyses [17]. $R^2$ represents the proportion of true heterogeneity that can be explained by the moderator; QM and its *P*-value determine the significance of the moderators in explaining heterogeneity; and QE and its *P*-value evaluate the significance of unexplained residual heterogeneity [23,24].

Funnel plots and Egger's test were employed to assess potential publication bias. Sensitivity analyses were conducted to evaluate the robustness of the pooled estimate [25,26]. Initially, outlier analyses were performed using Baujat plots. Studies located in the top right quadrant of the Baujat plot, or with studentized residuals exceeding 2 in absolute value, were considered potential outliers. After removing identified outliers, the overall pooled prevalence estimates were recalculated and compared with the main findings. Furthermore, we examined whether excluding smaller-sample data points (i.e., data points with the lowest quintile of sample sizes) yielded findings similar to the main results.

All statistical analyses were performed using R4.2.1 software (Lucent Technologies, Jasmine Mountain, USA). For all tests, p values less than 0.05 were considered statistically significant.

### Data collection on environmental factors and visualization of the spatial distribution and biogeographical characteristics

Baidu Map was used to determine the latitude and longitude coordinates of each study location. For human infection, all etiological confirmed paragonimiasis cases documented in population surveys, case reports, and case series were included in the spatial analyses. Environmental factors for each location, including annual mean temperature, annual precipitation, mean temperature of warmest quarter, precipitation of warmest quarter, mean temperature of coldest quarter, precipitation of coldest quarter were obtained from the WorldClim database (https://www.worldclim.org/) [27]. Altitude data was obtained from the Space Shuttle Radar Topography Mission (SRTM, http://www.gscloud.cn/) [28].

To visualize the spatial distribution of *P. westermani* and *P. skrjabini* infection, we georeferenced the etiologically definite human paragonimiasis cases and the prevalence of various animal hosts, and plotted them on a map of China using software ArcGIS10.7 (Environmental System Research Institute, Redlands, USA). The base layers of maps were downloaded from Resource and Environment Science Data Center of the Chinese Academy of Sciences (RESDC, http://www.resdc.cn) [29]. Additionally, scatter plots were used to illustrate the biogeographical characteristics of regions reporting *P. westermani* and *P. skrjabini* infection. T-tests were further conducted to explore the potential differences in biogeographical characteristics between the two *Paragonimus* species.

## Results

### Literature selection and quality assessment

Initially, 16,876 publications were identified through literature search. After removing duplicates, 10,642 articles were screened based on titles and abstracts, resulting in 1,880 articles for full-text assessment. Following full-text assessments, 38 studies were ultimately included in the meta-analysis for human infections, 107 for snail infections, 172 for infections in the second intermediate host, and 22 for infections in animal reservoirs (Fig 1). Additionally, we extracted human case report and case series from 965 publications (S1 Table). Among the collected literature, the earliest report of *Paragonimus* infection in humans or animals in China was in 1954 (S1–S5 Tables).

In the risk of bias assessment, all the studies were rated as having low to moderate bias (S2–S5 Tables). Specifically, 6 out of 38 publications for human infections, 81 out of 107 publications for snail infections, 143 out of 172 for the second intermediate hosts, and 20 out of 22 for animal reservoirs were rated as having low bias, the most common risk identified was the lack of random selection of the sample.

### Infection of *P. westermani* and *P. skrjabini* in humans

A total of 28,948 cases of human paragonimiasis have been reported in the literature, of which 2,401 cases occurred after 2010, 14,654 cases were male, and 6,089 cases were from rural areas (see Table 1). Additionally, a total of 10,076 cases of infection in children or adolescents have been reported, with 8,695 cases reported before 2010 and 1,381 reported after 2010. However, the number of cases by gender, source, and age is obviously higher than those documented, given that a considerable number is unspecified. As shown in Fig 2, human infections of *Paragonimus* have been documented in all provinces except for Tibet, Qinghai, Gansu, Ningxia, Macao, and Hong Kong. The cases of human infection are mainly documented in provinces or municipalities in the Yangtze River Basin, including Chongqing (6,035), Zhejiang (5,324), Hubei (4,945), Sichuan (2,896), and Hunan (1,414), which together account for 71.21% of the

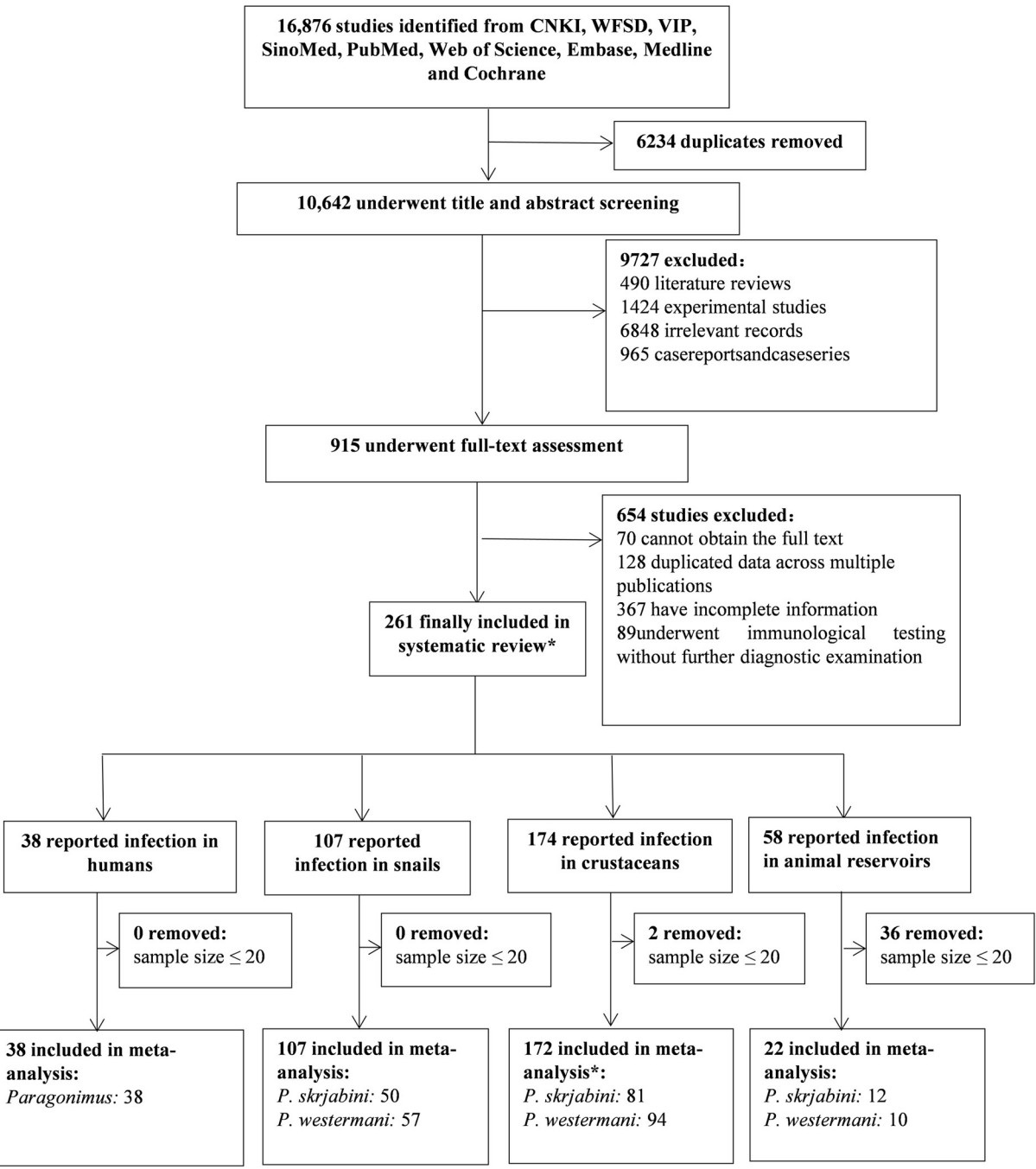

**Fig 1. Flow diagram of study selection for the systematic review and meta-analysis.** * Due to some studies simultaneously investigating humans, intermediate hosts, and reservoir hosts, the total number of literatures included in the systematic review (271) does not equal the sum of literature for different types of hosts. Similarly, several studies reported infection rates of both lung flukes in crustaceans, resulting in the total number of literature included in the study for the second intermediate host (172) being less than the sum of literature for *P. westermani* and *P. skrjabini*.

total national cases (see Table 1). It is worth noting that after 2010, there are still a considerable number of reported cases in areas such as Chongqing (1,073) and Sichuan (595), and many other provinces and municipalities also continue to report cases.

**Table 1. Characteristics of human paragonimiasis cases documented in China.**

| | 1954–1990 | 1990–1999 | 2000–2009 | 2010–present | Total |
|---|---|---|---|---|---|
| **Province** | | | | | |
| Chongqing | 779 | 984 | 3199 | 1073 | 6035 |
| Zhejiang | 1879 | 1649 | 1595 | 201 | 5324 |
| Hubei | 2466 | 989 | 1440 | 50 | 4945 |
| Sichuan | 640 | 537 | 1124 | 595 | 2896 |
| Guizhou | 1202 | 295 | 197 | 149 | 1843 |
| Hunan | 755 | 546 | 106 | 7 | 1414 |
| Shaanxi | 489 | 311 | 161 | 14 | 975 |
| Liaoning | 261 | 604 | 33 | 3 | 901 |
| Fujian | 490 | 33 | 341 | 4 | 868 |
| Anhui | 636 | 95 | 2 | 0 | 733 |
| Shanghai | 182 | 193 | 215 | 26 | 616 |
| Heilongjiang | 542 | 13 | 3 | 0 | 558 |
| Jiangsu | 82 | 369 | 38 | 40 | 529 |
| Henan | 176 | 104 | 44 | 64 | 388 |
| Beijing | 99 | 24 | 12 | 67 | 202 |
| Shandong | 0 | 176 | 5 | 0 | 181 |
| Jiangxi | 157 | 9 | 6 | 0 | 172 |
| Yunnan | 17 | 1 | 31 | 89 | 138 |
| Jilin | 92 | 3 | 0 | 2 | 97 |
| Guangdong | 4 | 23 | 19 | 15 | 61 |
| Shanxi | 0 | 14 | 14 | 0 | 28 |
| Guangxi | 1 | 23 | 1 | 0 | 25 |
| Hebei | 9 | 1 | 1 | 1 | 12 |
| Hainan | 0 | 0 | 3 | 0 | 3 |
| Inner Mongoria | 0 | 0 | 1 | 0 | 1 |
| Taiwan | 0 | 0 | 0 | 1 | 1 |
| Tianjing | 1 | 0 | 0 | 0 | 1 |
| Xinjiang | 0 | 1 | 0 | 0 | 1 |
| Gansu | 0 | 0 | 0 | 0 | 0 |
| Ningxia | 0 | 0 | 0 | 0 | 0 |
| Qinghai | 0 | 0 | 0 | 0 | 0 |
| Tibet | 0 | 0 | 0 | 0 | 0 |
| Hong Kong | 0 | 0 | 0 | 0 | 0 |
| Macao | 0 | 0 | 0 | 0 | 0 |
| **Age** | | | | | |
| < 18 | 2613 | 2660 | 3422 | 1381 | 10076 |
| ≥ 18 | 386 | 454 | 298 | 229 | 1367 |
| Not specified | 7960 | 3883 | 4871 | 791 | 17505 |
| **Gender** | | | | | |
| Male | 4201 | 4227 | 4587 | 1639 | 14654 |
| Female | 1677 | 1932 | 2220 | 677 | 6506 |
| Not specified | 5081 | 838 | 1784 | 85 | 7788 |
| **Source** | | | | | |
| Urban | 540 | 263 | 327 | 52 | 1182 |
| Rural | 1462 | 1270 | 2992 | 365 | 6089 |
| Not specified | 8957 | 5464 | 5272 | 1984 | 21677 |
| **Total** | 10959 | 6997 | 8591 | 2401 | 28948 |

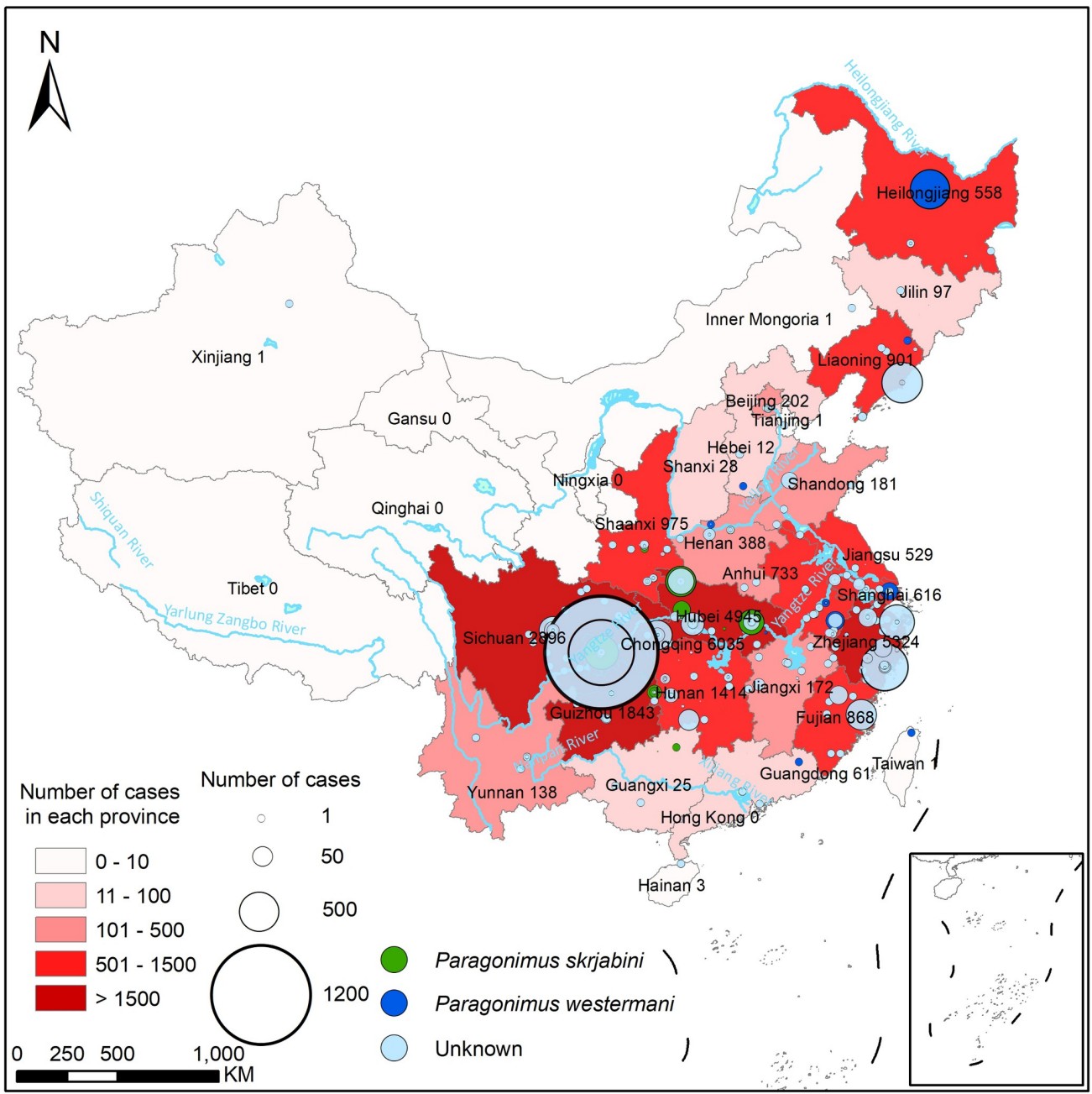

**Fig 2. Spatial distribution of human paragonimiasis cases documented in China.** The base layer of the map was downloaded from Resource and Environment Science Data Center of the Chinese Academy of Sciences (RESDC, http://www.resdc.cn).

Only a few cases differentiated whether the infection was caused by *P. westermani* and *P. skrjabini*. Cases of *P. westermani* infection are widely distributed, while *P. skrjabini* infections are primarily concentrated in more southern regions (Fig 2).

A total of 38 studies, containing 662,003 participants, reported screening for *Paragonimus* infection in human populations (see S2 Table), with 253 confirmed cases being reported and pooled prevalence of 0.05% (95% *CI*: 0.00–0.12%). The heterogeneity across the studies was high ($I^2$ = 93.3%, Table 2; forest plot shown in S1A Fig). Subgroup analysis and the meta-

**Table 2. Estimates of pooled prevalence and subgroup analysis of *Paragonimus* infection in humans.**

| | No. of data points | Sample size | No. of positive | Pooled prevalence, % (95% CI) | $I^2$, % | $R^2$, % (QM P value) | QE P value |
|---|---|---|---|---|---|---|---|
| **Pathogen** | **54** | **662003** | **253** | **0.05 (0.00; 0.12)** | **93.3** | **0.00 (0.575)** | **< 0.0001** |
| *P. westermani* | 30 | 58811 | 127 | 0.07 (0.00; 0.19) | 90.1 | | |
| *P. skrjabini* | 9 | 10636 | 22 | 0.04 (0.00; 0.22) | 87.1 | | |
| Not specified | 15 | 592556 | 104 | 0.03 (0.00; 0.16) | 95.2 | | |
| **Year of investigation** | | | | | | **4.36 (0.290)** | **< 0.0001** |
| 1954–1990 | 36 | 57540 | 184 | 0.08 (0.01; 0.20) | 88.4 | | |
| 1990–1999 | 7 | 32729 | 35 | 0.08 (0.00; 0.33) | 91.6 | | |
| 2000–2010 | 5 | 81900 | 25 | 0.02 (0.00; 0.21) | 66.7 | | |
| 2010–present | 6 | 489834 | 9 | 0.00 (0.00; 0.06) | 33.0 | | |
| **Gender** | | | | | | **4.17 (0.156)** | **< 0.0001** |
| Man | 8 | 16340 | 1 | 0.00 (0.00; 0.11) | 0 | | |
| Woman | 7 | 5174 | 0 | 0.00 (0.00; 0.13) | 0 | | |
| Not specified | 39 | 640489 | 252 | 0.09 (0.02; 0.19) | 95.2 | | |
| **Specimen** | | | | | | **0.00 (0.357)** | **< 0.0001** |
| Sputum | 38 | 100117 | 187 | 0.04 (0.00; 0.12) | 90.3 | | |
| Stool | 14 | 492277 | 22 | 0.03 (0.00; 0.18) | 79.2 | | |
| Stool or sputum | 2 | 69609 | 44 | 0.45 (0.04; 1.27) | 98.5 | | |

$R^2$ represents the proportion of true heterogeneity that can be explained by the moderator, the **QE** P value shows the significance of residual heterogeneity that is unaccounted for by the moderator, and the **QM** P value shows whether the moderator is statistically significant in explaining heterogeneity.

regression model indicated that none of the moderators could significantly explain the heterogeneity (see S6 Table).

## Infection of *P. westermani* and *P. skrjabini* in the first intermediate hosts

A total of 57 studies reported the presence of *P. westermani* infection in the first intermediate host (snails), with prevalence ranging from 0.00% to 6.72% (see S3 Table). The pooled prevalence of *P. westermani* in the first intermediate host was 0.11% (95% *CI*: 0.02–0.25%), and there was high heterogeneity across the studies ($I^2$ = 93.6%, Table 3; forest plot is presented in S1B Fig). *Semisulcospira* spp. was identified as the most common vector of *P. westermani*, with a pooled prevalence of 0.12% (95% *CI*: 0.02–0.28%). Additionally, *Tricula* spp., *Erhaiini* spp., and *Bythinella* spp. were identified as potential vectors of *P. westermani*.

Fifty studies reported *P. skrjabini* infection in the first intermediate host, with prevalence varied from 0.00% to 14.80% (see S3 Table). The pooled prevalence of *P. skrjabini* in the first intermediate host was 0.46% (95%*CI*: 0.27–0.70%), and the heterogeneity across studies was high ($I^2$ = 93.4%, Table 3; forest plot was shown in S1C Fig). The majority of infections in snails were reported in *Tricula* spp., with a pooled prevalence of 0.58% (95% *CI*: 0.28–0.96%). Additionally, *Pseudobythinella* spp., *Bythinella* spp., *Semisulcospira* spp., *Oncomelania* spp., *Erhaiini* spp., and *Akiyoshia* spp. were also potential vectors of *P. skrjabini*.

Spatial distribution of *P. westermani and P. skrjabini* infection in the first intermediate hosts is depicted in Figs 3 and 4. In the northeast area of China, *Semisulcospira* spp. serve as the primary transmission vectors of *Paragonimus*, and only *P. westermani* infection has been reported in this region. In more southern areas, *Semisulcospira* spp. are identified as the primary transmission vectors of *P. westermani*, while *Tricula* spp. are identified as the primary transmission vectors of *P. skrjabini*.

**Table 3. Estimates of pooled prevalence and subgroup analysis of *Paragonimus* infection in the first intermediate hosts.**

| | No. of data points | Sample size | No. of positive | Pooled prevalence, % (95% *CI*) | $I^2$, % | $R^2$, % (QM *P* value) | QE *P* value |
|---|---|---|---|---|---|---|---|
| ***P. westermani*** | **61** | **263423** | **639** | **0.11 (0.02; 0.25)** | **93.6** | | |
| **Year of investigation** | | | | | | **3.35 (0.149)** | **< 0.0001** |
| 1954–1990 | 32 | 120342 | 425 | 0.14 (0.01; 0.36) | 91.7 | | |
| 1990–1999 | 12 | 69290 | 52 | 0.04 (0.00; 0.27) | 89.4 | | |
| 2000–2009 | 11 | 58357 | 81 | 0.00 (0.00; 0.24) | 85.6 | | |
| 2010–present | 6 | 15434 | 81 | 0.62 (0.14; 1.39) | 97.3 | | |
| **Genus of snail** | | | | | | **0.00 (0.637)** | **< 0.0001** |
| *Semisulcospira* | 54 | 240158 | 599 | 0.12 (0.02; 0.28) | 94.1 | | |
| *Tricula* | 5 | 15918 | 11 | 0.04 (0.00; 0.41) | 62.5 | | |
| *Erhaiini* | 1 | 6227 | 26 | 0.42 (0.00; 2.49) | NE | | |
| *Bythinella* | 1 | 1120 | 3 | 0.27 (0.00; 2.27) | NE | | |
| ***P. skrjabini*** | **75** | **411797** | **1343** | **0.46 (0.27; 0.70)** | **93.4** | | |
| **Year of investigation** | | | | | | **0.00 (0.678)** | **< 0.0001** |
| 1954–1990 | 22 | 112904 | 428 | 0.52 (0.18; 1.01) | 92.6 | | |
| 1990–1999 | 14 | 75143 | 247 | 0.34 (0.02; 0.90) | 93.9 | | |
| 2000–2009 | 24 | 193038 | 502 | 0.34 (0.07; 0.75) | 91.9 | | |
| 2010–present | 15 | 30712 | 166 | 0.74 (0.26; 1.43) | 95.5 | | |
| **Genus of snail** | | | | | | **0.00 (0.830)** | **< 0.0001** |
| *Tricula* | 36 | 253031 | 643 | 0.58 (0.28; 0.96) | 94.6 | | |
| *Pseudobythinella* | 11 | 64914 | 340 | 0.57 (0.11; 1.32) | 90.5 | | |
| *Bythinella* | 10 | 9481 | 79 | 0.41 (0.01; 1.19) | 84.7 | | |
| *Semisulcospira* | 9 | 63806 | 178 | 0.06 (0.00; 0.60) | 79.6 | | |
| *Erhaiini* | 3 | 5789 | 26 | 0.80 (0.00; 2.66) | 92.4 | | |
| *Akiyoshia* | 3 | 2575 | 20 | 0.71 (0.00; 2.48) | 89.5 | | |
| *Oncomelania* | 2 | 10925 | 57 | 0.20 (0.00; 2.28) | 0.0 | | |
| *Assiminea* | 1 | 1276 | 0 | 0.00 (0.00; 1.83) | NE | | |

**NE**: not estimated; $R^2$ represents the proportion of true heterogeneity that can be explained by the moderator, the **QE** *P* value shows the significance of residual heterogeneity that is unaccounted for by the moderator, and the **QM** *P* value shows whether the moderator is statistically significant in explaining heterogeneity.

Subgroup analysis and the meta-regression model indicated that the prevalence of *P. westermani* and *P. skrjabini* in the first intermediate host did not exhibit significant differences across different snail genera and time periods (see Tables 3 and S7).

## Infection of *P. westermani* and *P. skrjabini* in the second intermediate hosts

In total, 94 studies reported *P. westermani* infection in the second intermediate host (see S4 Table), with a pooled prevalence of 52.02% (95% *CI*: 44.35–59.64%) and high heterogeneity across studies ($I^2$ = 99.6%, Table 4; forest plot presented in S1D Fig). Genus *Cambaroides* was identified as the primary second intermediate host for *P. westermani* in the northeastern areas of China (Fig 5), with a pooled prevalence of 59.79% (95% *CI*: 42.65–75.79%; Table 4). In other areas of China, *Sinopotamon* spp. were the primary second intermediate host, with a pooled prevalence of 52.86% (95% *CI*: 43.68–61.94%); other freshwater crabs such as *Nanhaipotamon* spp. and *Huananpotamon* spp. could also serve as the second intermediate host (see Tables 4 and S4).

Eighty-one studies reported *P. skrjabini* infection in the second intermediate host (see S4 Table), with a pooled prevalence of 30.37% (95% *CI*: 24.72–36.34%) and high heterogeneity across studies ($I^2$ = 99.8%, Table 4; forest plot presented in S1E Fig). In the northeastern region

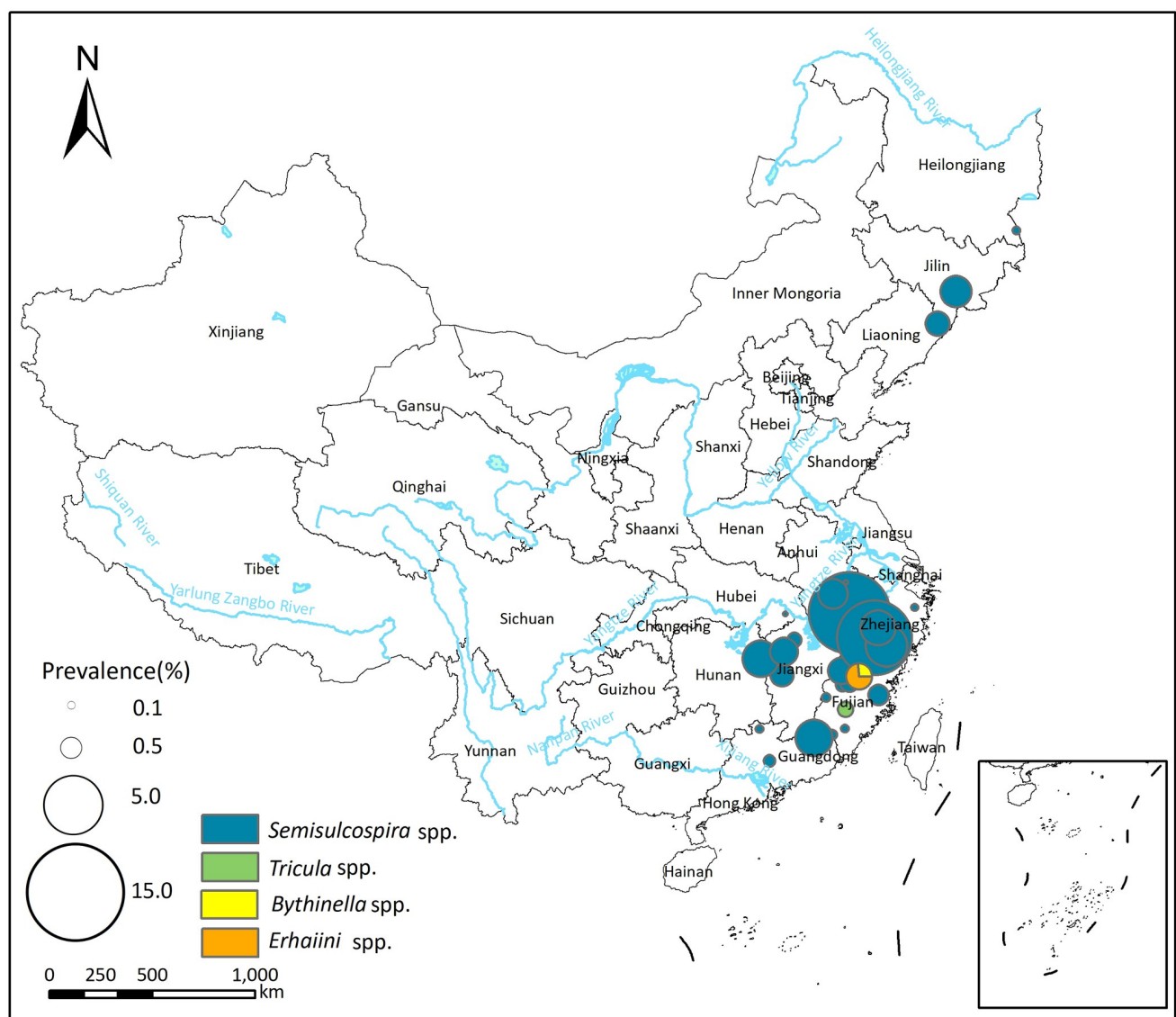

**Fig 3. Spatial distribution of *P. westermani* infection in the first intermediate hosts in China.** The base layer of the map was downloaded from Resource and Environment Science Data Center of the Chinese Academy of Sciences (RESDC, http://www.resdc.cn).

of China, only *P. westermani* has been reported in the second intermediate host, with no reports of the existence of *P. skrjabini* (see Fig 6). The second intermediate hosts of *P. skrjabini* included crabs of the Potamidae, Lithodidae, and Parathelphusidae families. Crabs of the Potamidae family were the most common second intermediate host, with *Sinopotamon* spp. being the most significant, exhibiting a pooled prevalence of 31.53% (95% *CI*: 24.92% - 38.53%). Additionally, other freshwater crabs such as *Nanhaipotamon* spp., *Potamon* spp., and *Tenuilapotamon* spp. of the Potamidae family, *Somanniathelphusa* spp. of the Parathelphusidae family, and *Malayopotamon* spp. of the Lithodidae family can also serve as the second intermediate hosts for *P. skrjabini* (see Table 4).

Subgroup analysis and the meta-regression model indicated that the prevalence of *P. westermani* and *P. skrjabini* in the second intermediate host did not exhibit significant differences

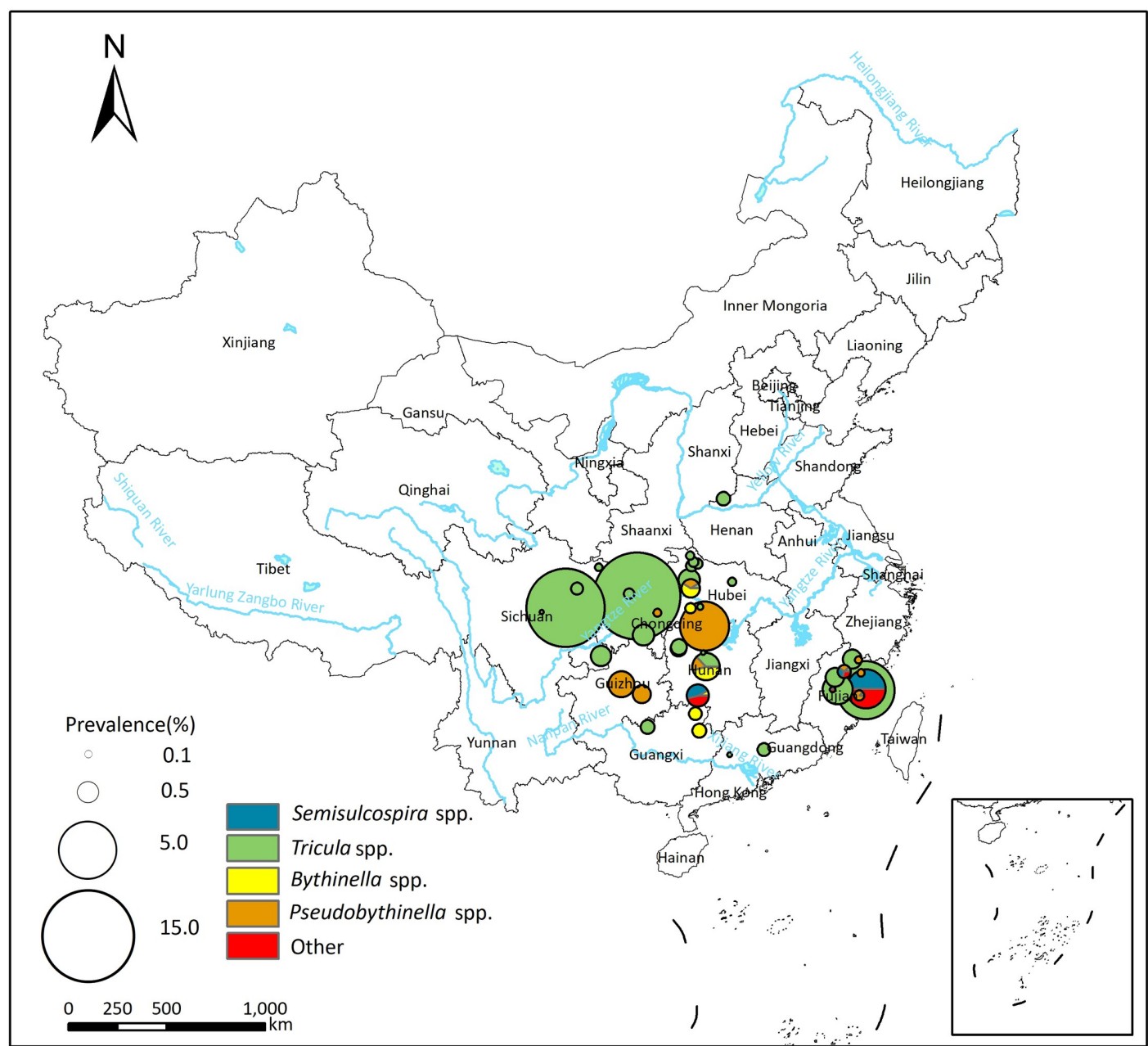

**Fig 4. Spatial distribution of and *P. skrjabini* infection in the first intermediate hosts in China.** The base layer of the map was downloaded from Resource and Environment Science Data Center of the Chinese Academy of Sciences (RESDC, http://www.resdc.cn).

among different crustacean genera, across different time periods, and with different detection methods (see Tables 4 and S8).

## Infection of *P. westermani* and *P. skrjabini* in animal reservoirs

Overall, 10 studies reported *P. westermani* infection in animal reservoirs (see S5 Table), with a pooled prevalence of 21.40% (95% *CI*: 7.82–38.99%) and high heterogeneity across studies ($I^2$ = 94.9%, Table 5; forest plot presented in S1F Fig). Cats (37.15% (95% *CI*: 9.61–69.92%)) and

**Table 4. Estimates of pooled prevalence and subgroup analysis of *Paragonimus* infection in the second intermediate hosts.**

| | No. of data points | Sample size | No. of positive | Pooled prevalence, % (95% *CI*) | $I^2$, % | $R^2$, % (QM *P* value) | QE *P* value |
|---|---|---|---|---|---|---|---|
| ***P. westermani*** | **100** | **165276** | **40049** | **52.02 (44.35; 59.64)** | **99.6** | | |
| **Year of investigation** | | | | | | 3.27 (0.097) | < 0.0001 |
| 1954–1990 | 44 | 86716 | 24212 | 62.67 (51.44; 73.25) | 99.5 | | |
| 1990–1999 | 15 | 66150 | 10488 | 43.24 (24.96; 62.51) | 99.8 | | |
| 2000–2009 | 22 | 6490 | 2762 | 41.69 (26.41; 57.82) | 98.8 | | |
| 2010–present | 19 | 5920 | 2587 | 45.80 (29.06; 63.03) | 99.5 | | |
| **Genus of hosts** | | | | | | 0.00 (0.492) | < 0.0001 |
| *Sinopotamon* | 70 | 157429 | 35929 | 52.86 (43.68; 61.94) | 99.7 | | |
| *Nanhaipotamon* | 3 | 175 | 53 | 26.02 (0.26; 69.67) | 95.0 | | |
| *Huananpotamon* | 2 | 1349 | 697 | 27.65 (0.00; 79.19) | 98.8 | | |
| *Malayopotamon* | 1 | 21 | 3 | 14.29 (0.00; 88.26) | NE | | |
| *Lithodes* | 1 | 72 | 61 | 84.72 (14.42; 100.00) | NE | | |
| *Eriocheir* | 1 | 85 | 16 | 18.82 (0.00; 88.53) | NE | | |
| *Cambaroides* | 20 | 5515 | 3110 | 59.79 (42.65; 75.79) | 99.4 | | |
| *Macrobrachium* | 2 | 630 | 180 | 28.57 (0.00; 79.70) | 0.00 | | |
| **Detection method** | | | | | | 0.00 (0.501) | < 0.0001 |
| Artificial digestion | 55 | 94460 | 26665 | 50.15 (39.81; 60.49) | 99.4 | | |
| Direct compression | 25 | 8923 | 4135 | 59.75 (44.42; 74.17) | 99.5 | | |
| Not specified | 20 | 61893 | 9249 | 47.42 (30.79; 64.34) | 99.7 | | |
| ***P. skrjabini*** | **109** | **198209** | **41426** | **30.37 (24.72; 36.34)** | **99.8** | | |
| **Year of investigation** | | | | | | 1.90 (0.184) | < 0.0001 |
| 1954–1990 | 24 | 21578 | 4833 | 32.76 (19.69; 47.33) | 99.1 | | |
| 1990–1999 | 22 | 84633 | 6773 | 21.85 (11.35; 34.57) | 99.8 | | |
| 2000–2009 | 23 | 66211 | 23849 | 40.59 (27.31; 54.59) | 99.9 | | |
| 2010–present | 40 | 25787 | 5971 | 30.13 (18.95; 42.62) | 99.2 | | |
| **Genus of hosts** | | | | | | 7.59 (0.065) | < 0.0001 |
| *Sinopotamon* | 74 | 167883 | 37566 | 31.53 (24.92; 38.53) | 99.8 | | |
| *Nanhaipotamon* | 7 | 1318 | 480 | 34.30 (13.74; 58.42) | 80.5 | | |
| *Potamon* | 5 | 1911 | 458 | 26.03 (6.31; 53.05) | 99.1 | | |
| *Tenuilapotamon* | 3 | 3195 | 2182 | 27.96 (3.52; 63.48) | 99.6 | | |
| *Aparapotamon* | 3 | 1880 | 307 | 15.66 (0.00; 48.85) | 78.0 | | |
| *Bottapotamon* | 3 | 189 | 127 | 75.72 (39.60; 98.48) | 96.1 | | |
| *Malayopotamon* | 2 | 104 | 62 | 42.94 (5.44; 85.95) | 95.8 | | |
| *Huananpotamon* | 2 | 82 | 27 | 33.89 (1.83; 78.46) | 0.0 | | |
| *Sinolapotamon* | 1 | 3596 | 6 | 0.17 (0.00; 38.11) | NE | | |
| *Tiwaripotamon* | 1 | 3898 | 2 | 0.05 (0.00; 36.44) | NE | | |
| *Neilupotamon* | 1 | 116 | 6 | 5.17 (0.00; 58.20) | NE | | |
| *Parvuspotamon* | 1 | 223 | 73 | 32.74 (0.00; 89.38) | NE | | |
| *Potamiscus* | 1 | 24 | 23 | 95.83 (38.43; 100.00) | NE | | |
| *Tenuipotamon* | 1 | 141 | 38 | 26.95 (0.00; 85.44) | NE | | |
| *Lithodes* | 3 | 13627 | 69 | 9.41 (0.00; 39.93) | 97.0 | | |
| *Somanniathelphusa* | 1 | 22 | 0 | 0.00 (0.00; 47.47) | NE | | |
| **Detection method** | | | | | | 1.75 (0.135) | < 0.0001 |
| Artificial digestion | 68 | 172632 | 35089 | 26.01 (19.38; 33.23) | 99.9 | | |
| Direct compression | 28 | 17125 | 4220 | 36.91 (25.41; 49.19) | 98.3 | | |
| Not specified | 13 | 8452 | 2117 | 40.59 (23.73; 58.66) | 99.2 | | |

**NE**: not estimated; $R^2$ represents the proportion of true heterogeneity that can be explained by the moderator, the **QE** *P* value shows the significance of residual heterogeneity that is unaccounted for by the moderator, and the **QM** *P* value shows whether the moderator is statistically significant in explaining heterogeneity.

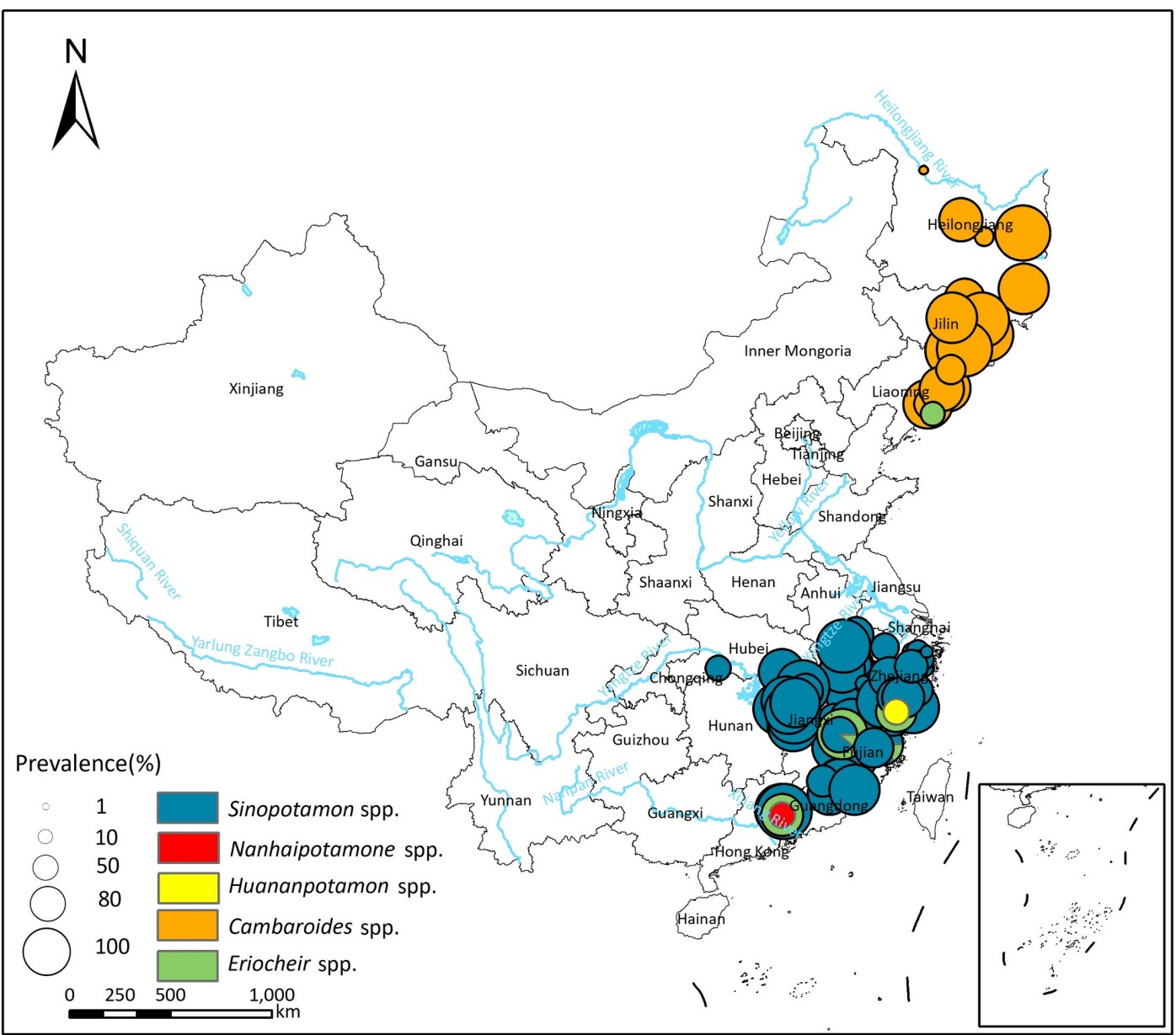

**Fig 5. Spatial distribution of *P. westermani* infection in the second intermediate hosts in China.** The base layer of the map was downloaded from Resource and Environment Science Data Center of the Chinese Academy of Sciences (RESDC, http://www.resdc.cn).

dogs (11.68% (95% *CI*: 0.00–36.56%)) were identified as the most common animal reservoirs for *P. westermani*.

Twelve studies reported *P. skrjabini* infection in animal reservoirs (see S5 Table), with a pooled prevalence of 20.31% (95% *CI*: 9.69–33.38%) and high heterogeneity across studies ($I^2$ = 95.2%, Table 5; forest plot presented in S1G Fig). Similar to *P. westermani*, cats (36.35% (95% *CI*: 20.74–53.51%)) and dogs (5.79% (95% *CI*: 0.00–23.03%)) were identified the most common animal reservoirs for *P. skrjabini*.

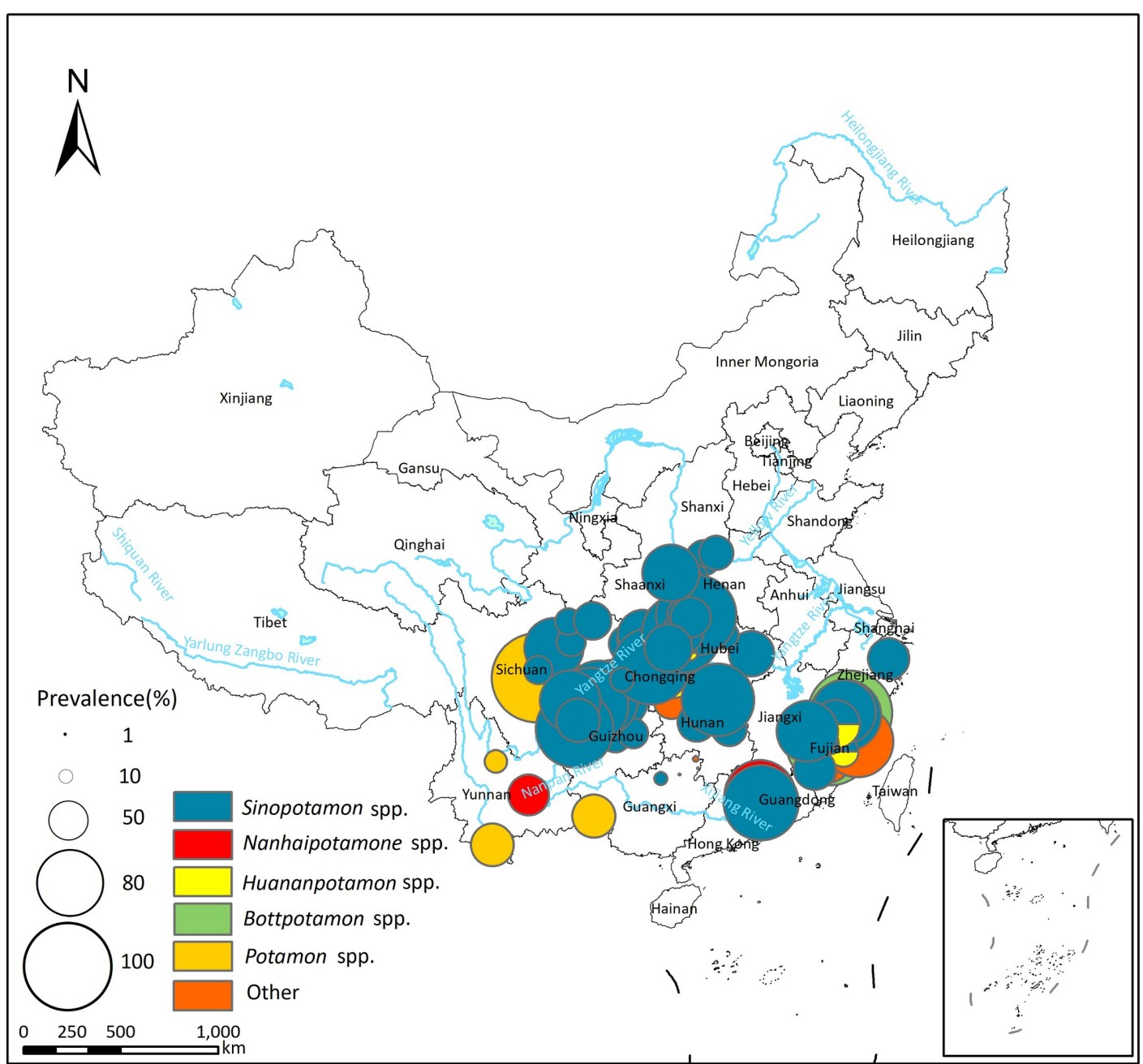

**Fig 6. Spatial distribution of *P. skrjabini* infection in the second intermediate hosts in China.** The base layer of the map was downloaded from Resource and Environment Science Data Center of the Chinese Academy of Sciences (RESDC, http://www.resdc.cn).

Subgroup analysis and meta-regression models indicated that animal categories, lifestyle (wild or domestic), or detection methods could significantly explain the observed heterogeneity (see Tables 5 and S9).

## Publish bias and sensitivity analysis

Asymmetry in the funnel plots and the results of Egger's test indicated the presence of publication bias (see S2 Fig). The sensitivity analysis results demonstrated that the pooled prevalence

**Table 5. Estimates of pooled prevalence and subgroup analysis of *Paragonimus* infection in animal reservoirs.**

| | No. of data points | Sample size | No. of positive | Pooled prevalence, % (95% *CI*) | *I²*, % | *R²*, % (QM *P* value) | QE *P* value |
|---|---|---|---|---|---|---|---|
| ***P. westermani*** | **13** | **1353** | **307** | **21.40 (7.82; 38.99)** | **94.9** | | |
| **Year of investigation** | | | | | | 5.40 (0.266) | < 0.0001 |
| 1954–1990 | 7 | 999 | 275 | 34.00 (12.82; 59.07) | 94.0 | | |
| 1990–1999 | 3 | 269 | 25 | 12.25 (0.00; 46.27) | 95.2 | | |
| 2010–present | 3 | 85 | 7 | 6.37 (0.00; 37.97) | 75.4 | | |
| **Family of hosts** | | | | | | 0.00 (0.609) | < 0.0001 |
| Canidae | 6 | 936 | 210 | 11.68 (0.00; 36.56) | 96.2 | | |
| Felidae | 5 | 299 | 74 | 37.15 (9.61; 69.92) | 96.1 | | |
| Viverridae | 1 | 66 | 13 | 19.70 (0.00; 85.85) | NE | | |
| Mustelidae | 1 | 52 | 10 | 19.23 (0.00; 85.87) | NE | | |
| **Life style** | | | | | | 0.00 (0.702) | < 0.0001 |
| Domestic | 10 | 1214 | 274 | 24.70 (4.93; 40.41) | 96.1 | | |
| Wild | 3 | 139 | 33 | 22.57 (1.18; 68.23) | 69.3 | | |
| **Detection method** | | | | | | 0.00 (0.545) | < 0.0001 |
| Sedimentation | 2 | 55 | 7 | 12.44 (0.00; 60.11) | 42.5 | | |
| Direct compression | 2 | 54 | 22 | 41.81 (2.41; 89.04) | 0.00 | | |
| Kato-Katz | 1 | 30 | 0 | 0.00 (0.00; 51.65) | NE | | |
| Not specified | 8 | 1214 | 278 | 23.59 (6.13; 47.41) | 96.6 | | |
| ***P. skrjabini*** | **20** | **1067** | **180** | **20.31 (9.69; 33.38)** | **95.2** | | |
| **Year of investigation** | | | | | | 10.34 (0.168) | <0.0001 |
| 1954–1990 | 5 | 199 | 53 | 30.53 (8.60; 58.19) | 94.5 | | |
| 1990–1999 | 5 | 408 | 17 | 3.52 (0.00; 21.03) | 91.3 | | |
| 2000–2009 | 5 | 167 | 56 | 30.31 (8.36; 58.09) | 83.4 | | |
| 2010–present | 5 | 293 | 54 | 23.88 (4.94; 50.39) | 96.3 | | |
| **Family of hosts** | | | | | | 26.53 (0.046) | < 0.0001 |
| Felidae | 11 | 433 | 146 | 36.35 (20.74; 53.51) | 93.7 | | |
| Canidae | 5 | 319 | 20 | 5.79 (0.00; 23.03) | 79.5 | | |
| Muridae | 1 | 223 | 0 | 0.00 (0.00; 29.15) | NE | | |
| Viverridae | 1 | 43 | 8 | 18.60 (0.00; 72.12) | NE | | |
| Suidae | 1 | 21 | 0 | 0.00 (0.00; 39.21) | NE | | |
| Mustelidae | 1 | 28 | 6 | 21.43 (0.00; 76.78) | NE | | |
| **Life style** | | | | | | 20.34 (0.018) | < 0.0001 |
| Domestic | 11 | 480 | 130 | 33.12 (17.50; 50.78) | 95.3 | | |
| Wild | 9 | 587 | 50 | 8.09 (0.40; 22.06) | 92.3 | | |
| **Detection method** | | | | | | 25.37 (0.029) | < 0.0001 |
| Direct compression | 6 | 231 | 90 | 45.69 (23.38; 68.90) | 92.7 | | |
| Sedimentation | 9 | 542 | 65 | 13.88 (3.16; 29.65) | 94.9 | | |
| Kato-Katz | 2 | 194 | 5 | 1.28 (0.00; 24.62) | 70.6 | | |
| Not specified | 3 | 100 | 20 | 15.93 (0.05; 46.32) | 89.7 | | |

**NE**: not estimated; *R²* represents the proportion of true heterogeneity that can be explained by the moderator, the **QE** *P* value shows the significance of residual heterogeneity that is unaccounted for by the moderator, and the **QM** *P* value shows whether the moderator is statistically significant in explaining heterogeneity.

estimate did not change significantly after the removal of outlier data points or data points with small sample sizes (95% *CI* overlapped; see S10 Table).

## Biogeographical characteristics of *P. westermani* and *P. skrjabini* infections

To investigate the biogeographical characteristics of *Paragonimus* occurrences, we created scatter plots using the climate features of *P. westermani* and *P. skrjabini* endemic sites and

1000 random points. The results indicate that, compared to random points, endemic sites of *P. westermani* and *P. skrjabini* are mainly distributed in regions with lower altitude and higher temperature and precipitation (see S11 Table and Fig 7). Specifically, endemic sites of *P. westermani* are predominantly distributed in areas with an altitude below 1166.0m, annual temperature above 1.0˚C, annual precipitation above 541.0mm, mean temperature of the warmest quarter above 18.3˚C, and precipitation of the warmest quarter above 304.0mm. On the other hand, endemic sites of *P. skrjabini* are distributed in areas with altitude below 2188.0m, annual temperature above 10.9˚C, annual precipitation above 578.0mm, mean temperature of the warmest quarter above 19.5˚C, and precipitation of the warmest quarter above 257.0mm. When comparing the two *Paragonimus* species, the endemic points of *P. westermani* have lower altitudes (below 1166.0m for *P. westermani*; 2188.0m for *P. skrjabini*) and lower mean temperature of the coldest quarter (above -20.1˚C for *P. westermani*; -0.8˚C for *P. skrjabini*).

## Discussion

In this study, we summarized the infection status and geographical distribution of *P. westermani* and *P. skrjabini* in humans and animal hosts in China. Our findings indicate that *Paragonimus* infection is widely distributed and remains prevalent in China, with variations in the transmission vectors, second intermediate hosts, and geographical distribution between *P. westermani* and *P. skrjabini*. Furthermore, environmental factors such as temperature and precipitation may influence the distribution of *Paragonimus*.

After years of educational efforts, the reported number of human paragonimiasis cases has significantly decreased in most areas of China (see Table 1). However, it is noteworthy that after 2010, a considerable number of reported cases persist in areas such as Chongqing (1073) and Sichuan (595), with many other provinces and municipalities also continuing to report cases, highlighting the need for ongoing control efforts against paragonimiasis. The number of cases in males is significantly higher than in females. This disparity is attributed to differences in behavior and occupational exposure. There is a higher proportion of males among fishermen, and males are more likely than females to catch and consume freshwater crabs and crayfish [30]. Another notable issue is the significant involvement of children and adolescents in paragonimiasis cases, both before and after 2010 [31–33]. In certain endemic areas, particularly in rural or mountainous regions, practices such as local children drinking untreated water and consuming undercooked shrimp and crab are more common among children compared to adults [34,35], underscoring the necessity for enhanced health education on paragonimiasis in schools in key areas. Additionally, human infection may be more widespread and underestimated due to a lack of training of health workers to identify paragonimiasis and a deficient case-reporting system [36].

The distribution regions of *P. westermani* and *P. skrjabini* in China exhibit both differences and overlaps. In the northeastern areas of China, only *P. westermani* has been documented, while in the southern part of China, both species coexist. The difference in the distribution of the two *Paragonimus* species is likely due to variations in their second intermediate hosts. Specifically, *Sinopotamon*, primarily distributed in the southern part of China, serves as the main second intermediate host for both *P. westermani* and *P. skrjabini* [37,38]. On the other hand, *Cambaroides*, which inhabits the northeastern region of China, can only serve as the second intermediate host for *P. westermani* [39]. On the other hand, *Cambaroides*, which inhabits the northeastern region of China, can only serve as the second intermediate host for *P. westermani* [30]. It has been reported that *P. westermani* and *P. skrjabini* share some common intermediate hosts, such as *Semisulcospira*, *Tricula*, *Erhaiini*, and *Bythinella* in the first intermediate host, and *Huananpotamon* in the second intermediate host [40,41]. Additionally, the cercariae

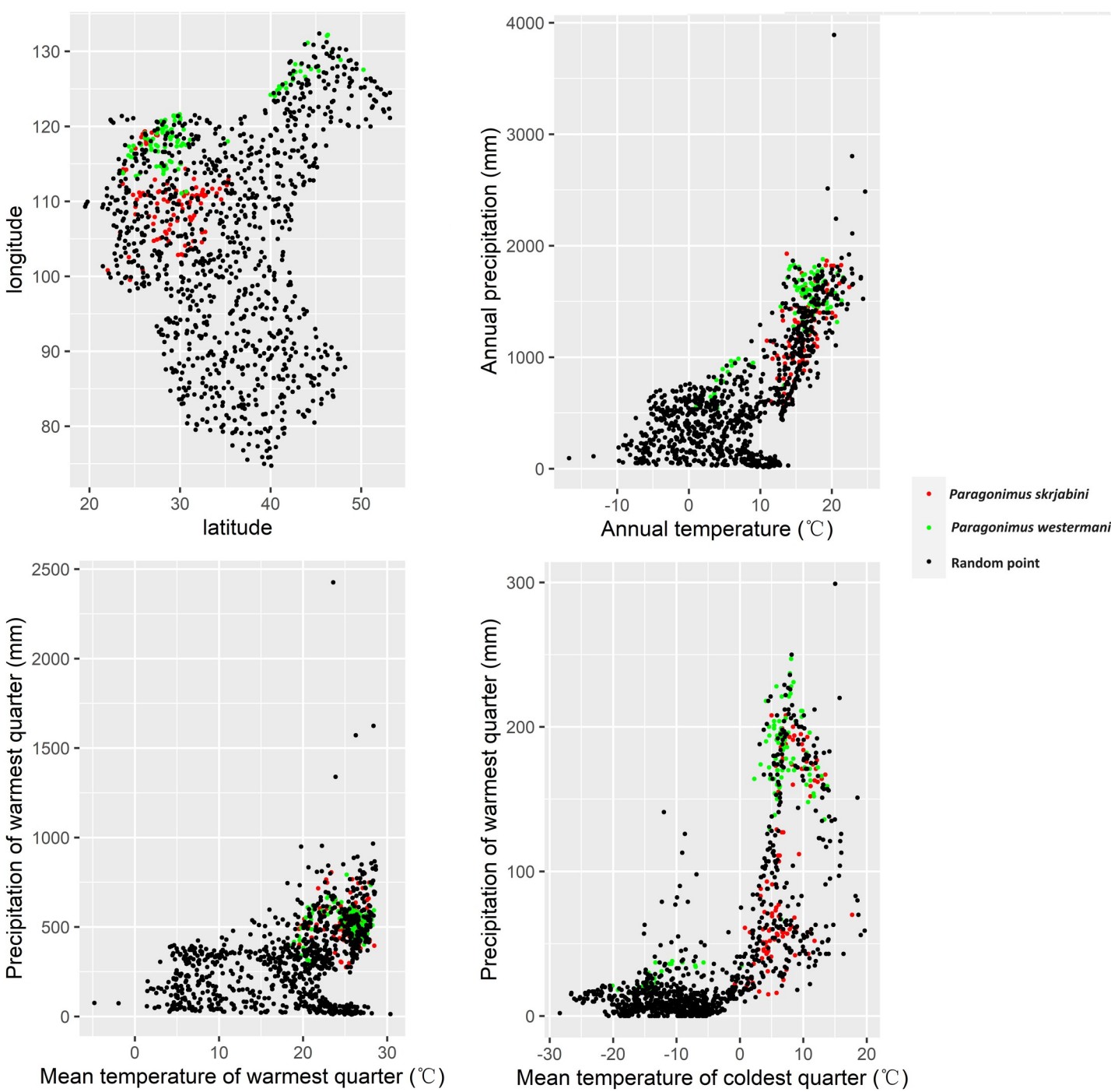

**Fig 7. Environmental characteristics of areas with reported *Paragonimus* infections in China.**

and metacercariae of *P. westermani* and *P. skrjabini* are morphologically similar [42]. Therefore, in areas where the two *Paragonimus* species overlap, there may be misclassification when detecting the infection in intermediate hosts. To accurately differentiate between the different *Paragonimus* species, nucleic acid detection is recommended to be conducted simultaneously in epidemiological surveys.

The prevalence of *Paragonimus* in intermediate hosts exhibits significant variation. In the first intermediate host, the prevalence of *P. westermani* ranged from 0.00% to 6.72%, while the prevalence of *P. skrjabini* ranged from 0.00% to 14.80% (see S3 Table). In the second intermediate host, the prevalence ranged from 0% to 100% (see S4 Table). None of the known moderators, including the taxonomic category of the intermediate host, year of survey, and detection methods, can significantly explain the heterogeneity across studies (see S7 and S8 Tables). Therefore, it is necessary to conduct random sampling surveys in different regions to further understand the factors that influence the prevalence of *Paragonimus* in intermediate hosts.

In many regions of China, it is common for residents to consume marinated or drunken crabs in their raw state [9,43]. However, the methods of salting and soaking in alcohol are not completely effective in killing the metacercariae [44,45]. Another prevalent practice is the consumption of freshwater crabs and crayfish through stir-frying, but inadequate heating may not fully eliminate the parasites [46,47]. Human infection occurs through the consumption of inadequately cooked freshwater crustaceans containing the infective metacercariae. Given the persistently high prevalence of *Paragonimus* in the second intermediate host (with a pooled prevalence of 52.02% (95% *CI*: 44.35–59.64%) for *P. westermani* and 30.37% (95% *CI*: 24.72–36.34%) for *P. skrjabini*; see Table 4), and the continued popularity of consuming raw or undercooked freshwater crustaceans in many areas of China, paragonimiasis remains a significant public health threat to the Chinese population.

The analysis of biogeographical characteristics revealed that temperature and precipitation might influence the distribution of *Paragonimus* (see Fig 7). Temperature may affect the distribution of *Paragonimus* by influencing the survival of the intermediate host (snails and crustaceans) or by affecting the development of *Paragonimus*. For example, research by Hu et al. indicates that the development of the eggs of *P. heterotremus* is closely related to the external temperature [48]. Development is slow or even halted at temperatures below 12˚C, and does not occur at temperatures above 37˚C. Chiu has found that the optimum temperature for the development of *P. iloktsuenensis* in *Tricula chiui* is 22 to 30˚C [49]. Our study results indicate that, compared to *P. skrjabini*, *P. westermani* can survive in regions with lower temperatures, such as northeastern China (see Figs 3–6), suggesting that *P. westermani* exhibits great tolerance to low temperatures. Similarly, Fan and colleagues have found that metacercariae of *P. westermani* can still develop into mature worms in rats after storage at 4˚C for up to 234 days [50].

*Paragonimus* infections have been predominantly documented in eastern China. This geographical distribution is closely associated with water supply, with precipitation playing a crucial role in the distribution of aquatic snails and crustaceans [51], both of which are integral to the *Paragonimus* life cycle. The higher levels of precipitation in eastern China create environments that are more conducive to the survival and proliferation of intermediate hosts, thereby increasing the risk of *Paragonimus* infections in these areas [52].

In this study, we pooled studies from numerous sites to achieve a relatively large sample size to summarize the prevalence of *P. westermani* and *P. skrjabini* in humans, intermediate hosts, and animal reservoirs. However, several limitations of our study should be considered. Firstly, the absence of literature reporting *Paragonimus* spp. infections in certain areas does not necessarily indicate that *Paragonimus* spp. infections do not exist there; it may be due to a lack of research in those areas or unpublished research findings. Secondly, significant heterogeneity was detected across studies, and most of the heterogeneity could not be explained by known moderators. Lastly, publication bias exists in this study, which may cause bias in the estimates of pooled prevalence. Therefore, the results of our study should be interpreted with caution. Despite these limitations, our study systematically summarizes the infection status of *P. westermani* and *P. skrjabini* in humans, intermediate hosts, and animal reservoirs in China, and elucidates their spatial distribution. The findings may provide valuable insights for the

control of paragonimiasis in China. In the future, it is advisable to incorporate paragonimiasis into China's National Notifiable Infectious Diseases Surveillance System to comprehensively monitor the incidence of the disease and identify high-risk populations more accurately. Furthermore, it is essential to systematically investigate the prevalence of *Paragonimus* spp. in various hosts in endemic areas and analyze the factors influencing these rates to enhance our understanding of the dynamics of *Paragonimus* spp. transmission.

## Conclusions

*Paragonimus* infection remains widely distributed and prevalent in China, with children and adolescents at high risk in endemic areas. Variations exist in the intermediate hosts and geographical distribution of *P. westermani* and *P. skrjabini* infections in China. *P. skrjabini* infections are predominantly concentrated in more southern regions compared to *P. westermani*. Additionally, altitude, temperature, and precipitation may influence the distribution of *P. westermani* and *P. skrjabini*.

## Supporting information

**S1 Table. Cases of human paragonimiasis docummented in literatures.**
(XLSX)

**S2 Table. Publications reporting *Paragonimus* prevalence in humans.**
(XLSX)

**S3 Table. Publications reporting *Paragonimus* prevalence in the first intermediate hosts.**
(XLSX)

**S4 Table. Publications reporting *Paragonimus* prevalence in the second intermediate hosts.**
(XLSX)

**S5 Table. Publications reporting *Paragonimus* prevalence in animal reservoirs.**
(XLSX)

**S6 Table. Multivariable meta-regression analyses for *Paragonimus* prevalence in humans.**
(XLSX)

**S7 Table. Multivariable meta-regression analyses for *Paragonimus* prevalence in the first intermediate hosts.**
(XLSX)

**S8 Table. Multivariable meta-regression analyses for *Paragonimus* prevalence in the second intermediate hosts.**
(XLSX)

**S9 Table. Multivariable meta-regression analyses for *Paragonimus* prevalence in animal reservoirs.**
(XLSX)

**S10 Table. Sensitivity analysis of the pooled prevalence of *Paragonimus* in humans, the first intermediate hosts, the second intermediate hosts, and animal reservoirs.**
(XLSX)

**S11 Table. Environmental characteristics of areas with reported *P. westermani* and *P. skrjabini* infections in China.**
(XLSX)

**S1 Fig. Forest plots of prevalence of *Paragonimus* species in humans, the first intermediate host, the second intermediate host, and animal reservoirs.** (a) *Paragonimus* in humans; (b) *P. westermani* in the first intermediate host; (c) *P. skrjabini* in the first intermediate host; (d) *P. westermani* in the second intermediate host; (e) *P. skrjabini* in the second intermediate host; (f) *P. westermani* in animal reservoir; (g) *P. skrjabini* in animal reservoir.
(DOC)

**S2 Fig.** Funnel plot for assessing publication bias in studies reporting prevalence of *Paragonimus* species in humans, the first intermediate host, the second intermediate host, and animal reservoirs (a) *Paragonimus* in humans; (b) *P. skrjabini* in the first intermediate host; (c) *P. westermani* in the first intermediate host; (d) *P. skrjabini* in the second intermediate host; (e) *P. westermani* in the second intermediate host; (f) *P. skrjabini* in animal reservoir; (g) *P. westermani* in animal reservoir.
(DOC)

## Author Contributions

**Conceptualization:** Fu-Yan Shi, Yan Lu, Lan-Hua Li.

**Data curation:** Kai Liu, Yuan-Chao Sun, Rui-Tai Pan, Ao-Long Xu, Han Xue.

**Formal analysis:** Kai Liu, Jin-Xin Zheng.

**Methodology:** Na Tian.

**Project administration:** Yan Lu, Lan-Hua Li.

**Software:** Kai Liu, Yuan-Chao Sun, Rui-Tai Pan.

**Supervision:** Fu-Yan Shi, Yan Lu, Lan-Hua Li.

**Validation:** Kai Liu, Yuan-Chao Sun, Rui-Tai Pan.

**Visualization:** Han Xue, Jin-Xin Zheng.

**Writing – original draft:** Kai Liu, Yuan-Chao Sun.

**Writing – review & editing:** Jin-Xin Zheng, Fu-Yan Shi, Yan Lu, Lan-Hua Li.

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
