## [Decision Letter · Decision Letter 0]

7 Jun 2024

Dear Dr. Li,

Thank you very much for submitting your manuscript "Infection and biogeographical characteristics of Paragonimus westermani and P. skrjabini in humans and animal hosts in China: a systematic review and meta-analysis" for consideration at PLOS Neglected Tropical Diseases. As with all papers reviewed by the journal, your manuscript was reviewed by members of the editorial board and by several independent reviewers. The reviewers appreciated the attention to an important topic. Based on the reviews, we are likely to accept this manuscript for publication, providing that you modify the manuscript according to the review recommendations. 

All reviewers concur on the study's value and quality, offering only minor comments. Kindly be aware that Reviewer 3's comments have been included as annotations in the PDF file.

Sincerely,

Qu Cheng, Ph.D.

Academic Editor

jong-Yil Chai

Section Editor

All reviewers concur on the study's value and quality, offering only minor comments. Kindly be aware that Reviewer 3's comments have been included as annotations in the PDF file.

Reviewer's Responses to Questions

**Key Review Criteria Required for Acceptance?**

**Methods**

-Are the objectives of the study clearly articulated with a clear testable hypothesis stated?

-Is the study design appropriate to address the stated objectives?

-Is the population clearly described and appropriate for the hypothesis being tested?

-Is the sample size sufficient to ensure adequate power to address the hypothesis being tested?

-Were correct statistical analysis used to support conclusions?

-Are there concerns about ethical or regulatory requirements being met?

Reviewer #1: Methodology and study design week articulated and appropriate.

Reviewer #2: The work complies with the aspects mentioned above. The period covered by the study, although the end date is stated (it is understood from 2010 to the present), the beginning could be better specified (it is a very broad period to say before 1990)

Reviewer #3: All in the MS

**Results**

-Does the analysis presented match the analysis plan?

-Are the results clearly and completely presented?

-Are the figures (Tables, Images) of sufficient quality for clarity?

Reviewer #1: The analysis presented do match the analysis plan and are well presented.

Reviewer #2: The work complies with the itms indicated above

Suggested details:

Line 202-203: The total number of publications is not clear (876 - 10642?)

Line 222-224: Table 1 does not show human cases in Hong Kong and Macau either.

Line 227: 71.12% or 71.21%

Fig 2: the reference system with parentheses is difficult to interpret

Line 219-222: it could be indicated that the cases by gender, source and age are obviously above those documented, explicitly consider that there is a significant number that is not specified

Fig 1: the value 261 is not clear? (38+107+174+58=377)

Linea 297: 94 estudios and line 307: 81 estudios = 175. En la linea 206 menciona 172 en second intermediate host

Reviewer #3: All in the MS

**Conclusions**

-Are the conclusions supported by the data presented?

-Are the limitations of analysis clearly described?

-Do the authors discuss how these data can be helpful to advance our understanding of the topic under study?

-Is public health relevance addressed?

Reviewer #1: Conclusions well justified and limitations well noted.

Reviewer #2: The items indicated above are fully contemplated in the work.

Reviewer #3: All in the MS

**Editorial and Data Presentation Modifications?**

Reviewer #1: No editorial suggestions other than what has been suggested to the authors below.

Reviewer #2: (No Response)

Reviewer #3: All in the MS

**Summary and General Comments**

Reviewer #1: This study provides a comprehensive meta-analysis for studies on the epidemiology paragonimiasis in China. I would suggest that the authors add in the Introduction some information about the possible socio-economic impacts of the disease in endemic areas. In the discussion: is there any explanation for the marked gender difference in the prevalence of the infection? Based on this extensive review I think it would be helpful to add any suggestions to standardize future surveys

Reviewer #2: The work is a valuable contribution by summarizing the situation of infection by P. westermani and P. skrjabini in their main hosts, as well as the geographical distribution. The figures that allow a visual-spatial representation of the infection situation in China stand out, being a fundamental pillar when establishing prevention-control measures.

Otros puntos:

When only Paragonimus is mentioned without specifying the species, it could be indicated as Paragonimus spp.

Line 87: P. westerman would be P. westermani

Line 87 - 91: would include some bibliographical reference

Line 446: ...prevalence of 52.02 % (95% CI 42.65 - 75.79%). The CI differs from Table 4

Line 458: P. iloktsuensis or P. iloktsuenensis?.

T. chiui: is it previously defined?

Reviewer #3: All in the MS

PLOS authors have the option to publish the peer review history of their article (what does this mean?). If published, this will include your full peer review and any attached files.

Reviewer #1: Yes: Ahmed Awad Adeel

Reviewer #2: Yes: Zully Hernández

Reviewer #3: Yes: José Alberto Iannacone Oliver

Figure Files:

Data Requirements:

Reproducibility:

References

---

## [Editor Report · Decision Letter 1]

9 Jul 2024

Dear Dr. Li,

We are pleased to inform you that your manuscript 'Infection and biogeographical characteristics of Paragonimus westermani and P. skrjabini in humans and animal hosts in China: a systematic review and meta-analysis' has been provisionally accepted for publication in PLOS Neglected Tropical Diseases.

Best regards,

Qu Cheng, Ph.D.

Academic Editor

Jong-Yil Chai

Section Editor

---

## [Editor Report · Acceptance letter]

30 Jul 2024

Dear Dr. Li,

We are delighted to inform you that your manuscript, "Infection and biogeographical characteristics of <i>Paragonimus westermani<i> and <i>P. skrjabini<i> in humans and animal hosts in China: a systematic review and meta-analysis," has been formally accepted for publication in PLOS Neglected Tropical Diseases.

Best regards,

Shaden Kamhawi

co-Editor-in-Chief

Paul Brindley

co-Editor-in-Chief
